# The structure of mass political belief systems: A network approach to understanding the left-right spectrum

Richard P. Bentall[1]*, Orestis Zavlis[2], Philip Hyland[3], Orla McBride[4], Kate Bennett[5], Todd K. Hartman[6]

1 Department of Psychology, University of Sheffield, United Kingdom, 2 Division of Psychology and Language Sciences, University College London, United Kingdom, 3 Department of Psychology, Maynooth University, Ireland, 4 School of Psychology, Ulster University, United Kingdom, 5 Department of Psychology, Liverpool University, United Kingdom, 6 Department of Social Statistics, University of Manchester, United Kingdom

* r.bentall@sheffield.ac.uk

## Abstract

Many socially consequential beliefs, notably political and religious ideologies, consist not of single propositions in isolation from others but as systems of many propositions. Philip Converse, one of the most influential political scientists of the twentieth century, proposed that such systems can be understood as networks of propositions and predicted that they would be highly intercorrelated in those with strong ideological commitments but less so in people who are less ideological. We used recent advances in network psychometrics to test this account in relation to the political beliefs of a representative sample of 2,058 UK adults, who rated themselves on the left-right dimension and then reported their attitudes toward 18 policy issues. We divided participants into equally-sized groups of left-wing, centrist and right-wing participants and found that, as Converse had predicted, the networks of those at either end of the left-right continuum were similar in structure, being significantly more interconnected than the networks of those who identified themselves as centrists, even though the actual beliefs were (for the most part) polar opposites. This finding, which was robust to sensitivity checks, aligns with previous research which has shown that people at the political extremes, compared to those in the centre, are more certain about their beliefs and less likely to change them over time. In each ideological group we also identified the same three communities of beliefs which mapped onto classic accounts of authoritarian attitudes, altruism and cooperativeness, and personal liberty. Attitudes towards gay rights had the highest predictability index in all three networks and was the most central node in the right and centre networks, suggesting that these attitudes play a largely unrecognised but important role in ideological positioning. Our analytical approach has implications for not only political beliefs but all organized belief systems.

**Data availability statement:** The full dataset used in the current study is publicly available at the Open Science Framework: https://osf.io/v2zur/.

**Funding:** The initial stages of the Covid 19 Psychological Research Consortium project were supported by start-up funds from the University of Sheffield (Department of Psychology, the Sheffield Methods Institute and the Higher Education Innovation Fund via an Impact Acceleration grant administered by the university) and by the Faculty of Life and Health Sciences at Ulster University. The research was subsequently supported by the UK Economic and Social Research Council under grant number ES/V004379/1 awarded to RPB, TKH, OMcB, and KB and others. The present analysis was further supported by a grant from Higher Education Innovation Fund awarded by the University of Manchester. The funders had no role in study design, data collection and analysis, decision to publish or preparation of the manuscript.

**Competing interests:** The authors have declared that no competing interests exist.

## Introduction

Many socially consequential belief systems consist of groups of propositions that are associated with each other, forming regular patterns that can be revealed by factor analysis. This is true, for example, of religious beliefs (in which beliefs about supernatural agents, the afterlife and moral imperatives in this life are often bundled together [1]) and conspiracy theories (in which beliefs about multiple conspiracies may express a general mistrust in institutions [2]) but it is especially obvious in the case of political beliefs, which are most commonly classified as left vs right-wing [3]. A standard interpretation of these patterns is that they reflect latent individual differences so that, for example, religious beliefs are attributed to over-sensitive agency detection [4], belief in conspiracy theories to a conspiracy mentality [5] and right-wing political beliefs to excessive cognitive and emotional rigidity [6].

In this paper, we focus on political beliefs although we will later argue that our findings are relevant to understanding belief systems in general. We begin by arguing that the conventional approach to understanding the left-right spectrum is too simple. After identifying some of its limitations, we apply recent advances in network analysis, also referred to as 'psychometric network analysis' [7,8], to explore the structure of political attitudes and test whether it varies as a function of location on the political spectrum. Our contribution to the research literature is a data-driven, exploratory method of mapping potential political belief systems without making strong assumptions about what should (or should not) be observed to reflect a coherent belief system. We constructed three network models of policy preferences among a large, nationally representative sample of the UK adult general population divided into those who classified themselves as being on the left, centre, and right on the political spectrum. Our network models confirm that citizens do indeed have interconnected political belief networks, and that they are stronger among those on the left and right (relative to those in the centre). Moreover, we discovered that political attitudes do cohere but that, as we anticipated might be the case, the network communities did not map onto the previously reported two factors of social and economic conservatism [9,10]; instead, they appeared to be constrained by three communities, which we discuss in more detail. We argue that our approach has implications for understanding belief systems in general and could be applied in domains other than politics.

### Ideology and the left-right continuum

One of the most enduring frameworks for categorizing people based on their political beliefs is the left-right spectrum, named after the seating arrangement in the National Assembly during the French Revolution (those who wanted change were seated to the left; those who favoured tradition, to the right). It has been argued that this categorization is a necessary and useful high-level abstraction and therefore that, "if the left-right distinction did not exist, scholars of ideology would need to invent its equivalent" [3] (p. 5). There is evidence that majorities of citizens across many different countries are comfortable locating their ideological commitments along a simple, left vs. right interval scale [11] and describe the distinction between the two poles in similar terms [12].

Some scholars have therefore argued that this left-right distinction is a durable feature of human nature and point to the fact that individuals who want to conserve institutions and maintain valued traditions, and those who wish to reform and change society, are recognisable in the politics of societies extending as far back as the Ancient Greeks [13]. It has been further argued that preference for these two poles reflects psychological characteristics such as (on the right) epistemic and existential needs that lead to cognitive rigidity [3,6]. Nonetheless, the characterization of this distinction has been the subject of controversy. Some scholars, for example, have argued that the tendency to construe ideology in terms of two poles reflects the didactic, adversarial nature of politics, so that the idea of a spectrum has persisted while its defining features may have changed over time, organized around attitudes towards traditional institutions in the past but, increasingly, as multiparty democracies have become the global norm, organized around attitudes towards inequality [14]. These two conceptions of what it means to be 'left' or 'right' may not be entirely conflicting as, historically, hierarchies and traditions have been construed as impediments to those who seek social and economic justice [15].

Nonetheless, the conventional way of construing the left-right spectrum faces three empirical challenges. The first is that factor analytic studies have rarely revealed a single dimension of belief corresponding to the spectrum. For example, some researchers have reported models in which there are separate dimensions reflecting social and economic conservatism versus liberalism [9,10]. Although typically highly correlated in the advanced economies of the west, when considered as predictors of preferences for specific policies these dimensions may sometimes align together but at other times may be in conflict [16]. A further complication is that, in the nations of Eastern Europe which, until 1989, lived under communist rule, socially conservative attitudes tend to be associated with left-wing economic attitudes [17] presumably because, in those countries prior to the fall of the Berlin Wall, redistributive, egalitarian policies had been the official doctrine of the governing elites and therefore represent the status quo.

In probably the largest cross-national study of political attitudes undertaken to date Malka and colleagues [18] used data from 325,804 people in 99 nations, collected as part of the World Attitude Survey. The researchers found that, globally, it was more common for socially conservative and economically liberal attitudes to be *negatively* correlated, albeit often to a small degree, than for them to be positively correlated, as in Western industrialised nations. Hence, the social and economic dimensions may have to be weighted differently when discriminating between right- and left-wing political parties depending on which countries or contexts are considered [19].

A second challenge is that, since the earliest days of research on political attitudes [20,21] it has been known that many citizens routinely lack coherent or ideologically constrained political attitudes [22] and instead rely on cognitive shortcuts when making their voting decisions [23,24]. For example, they may vote on the basis of a loosely felt sense of identifying with a particular party, or based on retrospective judgments of their well-being so that incumbents lose votes from voters who, for whatever reason, feel that they have not been flourishing [25].

Finally, there is some evidence that people who categorise themselves at either pole of left-right spectrum may be in some ways more similar to each other than to people who are non-ideological. For example, people who identify as belonging to either of the political extremes, compared to those who locate themselves in the centre of the spectrum, are much more likely to report certainty that their political beliefs are correct [26], are more likely to say that their beliefs are superior to those of other people, and are more likely to maintain their political beliefs over long periods of time [27]. Moreover, people who describe themselves as left or right-wing seem to be less influenced by contextual information than non-extremists (if asked to guess, say, the population of a large city like Chicago their estimates are less influenced by prior hints) [28].

## Political ideologies as belief systems

A seminal contribution to understanding the organization of political ideologies, which contrasted with the factor analytic approaches just reviewed, was made by the American social scientist Philip Converse [21]. His analysis (which, he argued, applied not only to ideologies but to any organized set of beliefs) began by defining a belief system as a network

or "configuration of ideas and attitudes in which the elements are bound together by some form of constraint or functional interdependence" (p.207).

Such a system has static constraints as evidenced by the fact that one attitude can very often be predicted given knowledge of others, but Converse also recognised the dynamic constraint that, over time, occurred when changes in some elements within the system would necessitate adjustment to others. Sometimes these constraints are logical (e.g., it is illogical to insist that government expenditure be increased, revenues are decreased, and the budget be balanced at the same time). However, the observed pattern of association between beliefs within a system often cannot be explained in this way (e.g., there is no logical reason why, in countries in which social and economic attitudes are aligned, people who support more spending for the armed forces also, typically, support the use of capital punishment). In these circumstances, constraints might be due to some hidden psychological processes that influence both kinds of beliefs (presumably authoritarian traits in the example just given) or the consequence of the way that ideas have been packaged together prior to dissemination by influential belief entrepreneurs (e.g., political theorists and their activist followers).

An innovative concept introduced by Converse was *centrality* (e.g., see pp.208–209): some beliefs within a system are more central than others, in the sense that a perturbation of these beliefs will require more radical adjustment of the system (imagine a Marxist deciding that class struggle is a relatively minor force in human history), compared with others that are more peripheral. Thus, Converse's insight was that the structure of belief systems would depend on the position of an individual within a 'vertical hierarchy' down which political information flows, with opinion-forming political elites at the apex, activists and committed party members below them, and 'mass publics' at the base. Elites and activists often spend many years discussing and refining their belief systems, so that the individual elements are understood in detail, the relationships between them tightly specified, and contradictions within the system either eliminated or reasoned away with exculpatory hypotheses. However, ordinary citizens may be much less reflective about their political commitments and, as a consequence their belief systems are much less constrained.

Converse supported this account with detailed analyses of qualitative data obtained from large numbers of American voters in the 1950s and early 1960s. His analyses identified five strata of voters: ideologues (estimated to be 3.5% of voters); near-ideologues who could relate policies to abstract political concepts (12%); those whose political commitments were aligned with specific group interests (45%); those who evaluated parties by temporal association (e.g., in the UK, someone might associate the Labour Party with the 2008 financial crisis) or because of gratitude for or dislike of specific policies (22%); and, finally, those who had no policy interests whatsoever but were simply loyal to a particular party or liked a particular candidate (17%). Importantly, in this scheme, only a fraction of voters (less than 20%) structured their political commitments primarily according to any kind of political philosophy. In these voters, compared with those in the third to fifth tiers, attitudes towards policies were highly correlated, and these more ideologically constrained citizens were much less likely to change their vote between one election and another.

## Network models of political belief systems

Although Converse's account has been considered a seminal contribution to the analysis of ideologies, the factor analytic methods which have been available to investigate the structure of political beliefs have been ill-suited for testing and elaborating his model. Nonetheless, some more recent observations made by political scientists are consistent with his original theory. For example, Feldman and Johnston's [10] observation that attitudes towards social and economic conservative policies are most highly correlated at both ends of the political spectrum but much less so at the centre is entirely consistent with Converse's original theory. Similarly, Baldassari and Goldberg [29] used a novel technique, relational class analysis, to identify groups of American voters who were similar in their organization of attitudes towards economic issues, civil rights, morality and foreign policy (the correlations between the issues were similar, even though their position with respect to each of these domains might be different) and identified three classes: *ideologues* (33%) whose beliefs corresponded to the traditional left-right dimension, *alternatives* (40%) whose attitudes towards economic issues contrasted

with their attitudes towards social issues (right economic while left social or vice versa), and *agnostics* (27%) whose beliefs in these different domains were only weakly associated with each other. The researchers found that the same pattern could be found over twenty years of the American National Election Survey (even though different questions were used and different participants were recruited in the different waves).

Network theory methodologies, which have their origins in physics [30] and have recently been used in the study of psychiatric disorders [31], are uniquely suited to pursuing Converse's agenda further, so much so that the coincidental alignment of these methods with his theoretical approach seems remarkable. These methodologies abandon the assumption that beliefs (or psychiatric symptoms) necessarily cluster together into recognizable ideologies (syndromes) because of common underlying causes such as specific cognitive traits (or underlying disease processes) but instead assume that the relationships between them emerge as a consequence of the way they can influence each other [31]. In respect to belief systems, this is easiest to see when there is some kind of logical relationship between the constituent propositions. For example, support for tax cuts implies support for limiting benefits and not for spending more money on defence, so that changing one of these attitudes will likely lead to changes in the others. However, there is no assumption that the relationships between attitudes are always logical in this way.

New analytical techniques allow the visual representation of networks, revealing putative interrelationships between the elements. The nodes in these visualisations correspond to the elements that have inter-relationships which we are trying to understand – in our case, attitudes towards particular political policies and ideas (in psychopathology they would be symptoms). The 'edges' or lines connecting the elements in the network represent the magnitude of the influence between them [32]. For example, within a specific network it may be that some nodes (attitudes, say towards multiculturalism and gay rights) are highly associated with each other (so that people who favour one will usually favour the other) whereas others (say between benefits for poor and international cooperation) are less so. Once networks have been specified, it is possible to identify communities of nodes that are highly interconnected and therefore ideologically constrained (changing one of the attitudes would very likely require changes in the others). Various centrality statistics are also available to estimate which nodes are most influential within the network as a whole by virtue of their connections with other nodes (in network analyses, the concept of 'centrality' exactly corresponds to Converse's use of the term more than sixty years ago). When networks are estimated for different subpopulations, network comparison statistics are available to test each network's overall connectivity, as well as the strength of individual edges for different groups. Hence, it is possible to determine whether different groups of citizens differ in the extent to which their networks are ideologically constrained.

Although the majority of psychological applications of network theory have thus far been in the field of mental health, these analytical techniques have been used to address psychological and social problems. For example, Causal Attitude Network (CAN) models, in which behavioural outcomes are seen as products of the interactions between a range of attitudinal and emotional factors connected in a network, have been used to explain the notoriously difficult to predict relationship between attitudes and actions. Dalege et al. [33] used a CAN model to study voters' decisions to support specific candidates in US elections between 1980 and 2012. Attitudes were found to better predict voting behaviour when the elements in the network were highly interconnected (e.g., during the 2012 election of Obama) than when the relationship between the elements was weak (e.g., the election of G.H.W. Bush in 1992). Moreover, in each election the most central node (e.g., the belief that Clinton would show strong leadership in 1992) was the strongest single predictor of voting behaviour. In a subsequent study, the same researchers [34] showed that American voters who reported being highly interested in politics had more highly connected networks than those who did not, and that connectivity of their networks predicted not only voting behaviour but stability of their attitudes over time.

Similarly, Boutyline [35] constructed network models of American voters' attitudes during the 2008 Presidential Election, including ideology (self-identification as liberal vs conservative) within the networks. The structure of the networks did not vary substantially between demographic groups, and ideology was the central node, a finding that is consistent with previous demonstrations that many ordinary people do organise their political beliefs according to the left-right distinction

[11,12]. Voters who were classified as 'high information' (e.g., better educated) had more connected networks than those classified as 'low information' (e.g., less educated), as predicted by Converse's account. Similarly, Brandt and his colleagues [36] studied panel data from New Zealand voters collected between 2008 and 2015, including symbolic (support for particular parties or identification as left vs right) and operational (policy) beliefs in their analyses, revealing that, across multiple years, symbolic nodes were more central in the networks.

In sum, previous research has examined the relationships between ideological and political variables by including them all as elements in a single network. Here, however, we attempt to assess how ideology *moderates* the relations between policy preferences across a range of different issues. To achieve this goal, it is necessary to examine variations in political attitudes among those whose ideological commitments are determined independently of the elements included in their networks. In the present study, we thus report a psychometric network analysis of policy preferences in a representative sample of the UK public in late 2021. Converse's model [21] leads us to predict that networks of policy attitudes should be most highly connected in those with *either* strong right- or left-wing commitments. We tested this prediction by comparing those who described themselves as aligned with the left or right with those who described themselves as belonging to the centre of the left-right political spectrum. We expected the networks to be not only highly interconnected at the extreme ends (left and right) but to also differ in other local and global features. Therefore, we also compared the networks in several other aspects: centrality and predictability indices, as well as their global community (or factor) structure. Finally, as our dataset included childrearing questions which have previously been shown to predict authoritarian predispositions [37–39], we used the opportunity to assess the external validity of our network community structure against this measure.

## Materials and methods

### Sample

Data come from the sixth wave of the Covid-19 Psychological Research Consortium longitudinal survey, which tracks the effect of the coronavirus pandemic on the mental health, attitudes, and behaviour of the UK population [40]. Data for the first wave of the survey were collected in March, 2020, when strict lockdown measures were announced. Quota sampling was used to recruit 2,025 adults (i.e., those over the age of 18 years old) stratified by age, sex, and household income to be representative of the UK population; validation checks conducted at that time found that the sample was also broadly representative on many other variables, including diversity of political allegiances as judged by self-reported voting in the 2016 referendum on Britain's exit from the European Union, and also in the 2019 UK General Election. Subsequent waves re-contacted participants from earlier waves and then recruited further respondents to replace those lost to panel attrition. As a consequence, subsequent waves were slightly less representative than the first (but still quite diverse).

Wave 6 data were collected between 6th of August and 28th of September 2021 and consisted of 1,643 people who had completed one or more earlier survey waves, and an additional 415 newly recruited participants (*n* = 2,058). Females (*n* = 1,069) were slightly over-represented compared to males (983; 4 transgender, 2 prefer not to say). Participants came from across the four constituent administrations of the UK (1,702 from England, 205 from Scotland, 108 from Wales and 43 from Northern Ireland; these numbers conform closely to the proportions of the UK population reported by the UK Office of National Statistics of 84.4%, 8.05%, 4.5% and 2.8% respectively [41]). 42.7 percent were university graduates (as opposed to 33.8% with a graduate-level qualification according to the 2021 UK census of England and Wales). When asked about their social class, 42.3% described themselves as working class, 39.6% as middle class, 1.3% as upper class, and 16.9% said they did not belong to a class. Of the 69.7 percent who reported voting in the 2016 referendum on Britain leaving the European Union, 41.8 percent reported voting leave, so remain voters were over-represented (58.2 percent); no data on voting in the 2019 general election was available from this wave. Ethical approval for the study was granted by a blinded for review process (University of Sheffield ethics ref no. 033759). The full dataset used in the current study is publicly available at the Open Science Framework: https://osf.io/v2zur/.

## Measures

There was no missing data because our survey software required responses for every item before moving on.

**Political attitudes.** Participants reported their preferences toward eighteen political issues, using a scale modelled on Wilson's [42] questionnaire measuring right- vs left-wing attitudes. The issues were chosen to reflect potential social and economic dimensions of ideology that are relevant to current times; these were: Support for "the death penalty"; "spending money on the armed forces"; "multiculturalism"; "stiff jail terms for criminals"; "euthanasia (physician assisted suicide)"; "gay rights"; "higher benefits for the poor"; "immigration"; "legalised abortion"; "lower taxes to promote business"; "international government"; "rehabilitation of offenders"; "traditional family values"; "monogamy"; "trans rights"; "Brexit"; "public demonstrations such as 'taking the knee' to acknowledge discrimination against ethnic minorities"; and, finally, "international aid." Support for each policy was recorded on a five-point Likert scale (1 = "strongly disagree"; 5 = "strongly agree"). The validity of the items is established by their relationship with self-identification on the left-right spectrum, shown in the results, Table 1. It is important to note that the network psychometric approach taken in this study does not make the assumption that all of these items form a scale; nonetheless, Cronbach's alpha for the items (with items rescored so that high scores always reflect the choices made by self-defined conservatives as shown in Table 1) was .83.

**Left-right orientation.** A single item taken from the British Election Study [43] asked participants how they would describe their own political ideology (on a 10-point scale ranging from 1 "left-wing" to 10 "right-wing"). This approach to self-identifying location on the spectrum has been employed in many studies of political ideology (e.g., [3,26]).

**Child-Rearing preferences.** Child-rearing preferences have long been used to measure authoritarian personality traits in a way that avoids explicitly inquiring about political beliefs [38,39]. Studies show that these questions can be used to predict political choices [44] and correlate highly with, and are less sensitive to circumstances, than conventional political

**Table 1. Descriptive statistics and item analyses for individual policy attitudes.**

|  | Mean (Standard Deviation) | | | Test Statistic, *p* Value | | |
| --- | --- | --- | --- | --- | --- | --- |
| Variable | Left | Centre | Right | Centre – Left | Centre – Right | Left – Right |
| P1 (Death Penalty) | 1.94 (1.24) | 3.09 (1.33) | 3.51 (1.32) | 10871, <.001* | 66428, <.001* | 9045, <.001* |
| P2 (Spending money on Army) | 2.86 (1.16) | 3.61 (0.98) | 3.96 (0.92) | 14713, <.001* | 56564, <.001* | 9087, <.001* |
| P3 (Multiculturalism) | 4.03 (1.00) | 3.36 (0.98) | 3.19 (1.17) | 57831, <.001* | 40706,.05 | 62606, <.001* |
| P4 (Stiff Jail Terms) | 3.36 (1.13) | 3.97 (0.95) | 4.09 (0.94) | 16739, <.001* | 46481,.11 | 16127, <.001* |
| P5 (Euthanasia) | 3.72 (1.08) | 3.52 (1.07) | 3.59 (1.15) | 38090, <.01 | 49499,.32 | 34121, <.001* |
| P6 (Gay Rights) | 4.38 (0.94) | 3.68 (1.10) | 3.48 (1.19) | 47592, <.001* | 42066, <.01 | 58480, <.001* |
| P7 (Higher Benefits for the Poor) | 4.02 (0.93) | 3.33 (1.01) | 3.10 (1.19) | 57607, <.001* | 41180, <.001* | 66469, <.001* |
| P8 (Immigration) | 3.82 (1.06) | 3.02 (1.08) | 2.84 (1.29) | 60478, <.001* | 47861,.09 | 67261, <.001* |
| P9 (Legalized Abortion) | 4.24 (1.01) | 3.66 (1.03) | 3.74 (1.12) | 45371, <.001* | 45010,.29 | 44670, <.001* |
| P10 (Lower Taxes) | 3.04 (1.05) | 3.42 (0.84) | 3.63 (0.94) | 20848, <.001* | 57014, <.001* | 16594, <.001* |
| P11 (International Government) | 3.30 (0.09) | 3.00 (0.84) | 3.08 (1.13) | 29139, <.001* | 43551,.04 | 39920, <.001* |
| P12 (Rehabilitation of Offenders) | 4.00 (0.92) | 3.47 (0.92) | 3.45 (1.04) | 46583, <.001* | 45441,.98 | 53207, <.001* |
| P13 (Traditional Family Values) | 3.17 (1.15) | 3.86 (0.92) | 4.13 (0.92) | 14059, <.001* | 53919, <.001* | 12791, <.001* |
| P14 (Monogamy) | 3.52 (1.05) | 3.58 (0.98) | 3.86 (1.04) | 27693,.08 | 52088, <.001* | 24575, <.001* |
| P15 (Trans Rights) | 4.02 (1.05) | 3.37 (1.08) | 3.13 (1.22) | 55107, <.001* | 36285, <.01 | 64888, <.001* |
| P16 (Brexit) | 1.91 (1.26) | 3.07 (1.26) | 3.76 (1.24) | 12628, <.001* | 81112, <.001* | 6785, <.001* |
| P17 (Public Demonstrations) | 3.87 (1.11) | 3.00 (1.12) | 2.72 (1.36) | 64308, <.001* | 45963,.01 | 74293, <.001* |
| P18 (International Aid) | 4.00 (1.05) | 3.15 (1.04) | 3.04 (1.25) | 60792, <.001* | 44439,.40 | 67950, <.001* |

Responses for each policy item were recorded on a five-point Likert scale, where 1 = strongly disagree and 5 = strongly agree. Test Statistic (W) is from the Wilcoxon rank sum test; * indicates that test remained statistically significant after Bonferroni correction.

attitude measures [37]. Here we used the items introduced into the American National Election Study and then used by Feldman and Stenner [38]. They consisted of four forced-choice binary questions to "identify the qualities that people feel children should have": *independence* vs. *respect for elders*, *curiosity* vs. *good manners*, *obedience* vs. *self-reliance*, and *being considerate* vs. *well-behaved*. Items corresponding to higher levels of authoritarianism were coded as 1 (i.e., respect for elders, good manners, obedience, and well-behaved); while others reflecting the opposite were coded as 0 (i.e., independence, curiosity, self-reliance, and being considerate). These items were then summed into a total score for authoritarianism. In this study, the reliability of the scale was marginal, alpha = .58, but the scale was not used in our primary analyses and was employed solely for the purpose of validating our community structure (see below).

## Analysis plan

Our analyses were conducted using the statistical software R (version 4.1.2). Listwise deletion was employed to handle missing values. Participants were divided into three groups based on their self-reported position on the left-right spectrum (ranging from 1 "Left" to 10 "Right"), which was approximately normally distributed. Respondents with scores of 3 or less were assigned to the left-wing group; those with scores of 8 or above were assigned to the right-wing group; and the rest (that is, those who scored 4, 5, 6, or 7) were classified as centrists. Given the ordinal nature of belief responses, the Wilcoxon rank sum test was used to assess (median) differences in their endorsement across the political groups (which were expected for most, if not all, of the political beliefs). In order to correct for multiple testing (here as well as elsewhere), Bonferroni correction was employed.

**Network psychometrics.** We estimated three network models, one for each political group (i.e., networks of left, centre, and right), and then examined them for similarities and differences. Although our local network comparisons were exploratory, the global ones were hypothesis-driven. In particular, the global strength of networks (that is, the absolute level of their connectivity) was expected to be significantly stronger in the networks at the extreme ends of the political spectrum (i.e., left and right). Conversely, although no particular predictions were made for the network's community structure, we were interested to examine whether the emerging latent factors would correspond to previous factor-analytic findings of, for example, separate social and economic dimensions of the left-right spectrum [9,10].

**Network estimation.** Given the ordinal nature of our data, the Gaussian Graphical Model (GGM) was fitted for each ideological group [45]; previous research has shown that five point interval scales such as those used in this study can be safely treated as continuous variables and are therefore are suitable for GGM analyses [46]. In graphical (network) models, political issues are represented by nodes, and the associations between them by edges (connections). Our GGMs were estimated using the common estimation procedure of graphical least absolute shrinkage and selector operator (or glasso), in conjunction with model selection through the Extended Bayesian Information Criterion (EBIC) (with the hyperparameter, gamma, set to 0.5 in order to ensure sparser models). To aid interpretability, the three graphs were visualized in the same manner, using the average layout (across the three networks), as determined via the Fruchterman-Reingold algorithm [47]. Following recent recommendations [48], a replication of our networks was conducted on the same data and is reported in the Supplement. The estimation and visualization of our networks was conducted through the R package 'qgraph' [49].

**Network comparisons.** Four network properties were contrasted across our three political networks, of which two were global: (1) *Global Strength* (which is the sum of absolute edge-weight values within a given network); (2) *Community* (or factor) *structure* (which is the set of reflective latent variables that summarize the clustering of nodes); the other two were local: (3) *Strength Centrality* (that is, the sum of absolute connectivity that each node exhibits within a network); and (4) *Predictability* (i.e., amount of variation explained in a given node).

Two of these properties were statistically compared using the Network Comparison Test (NCT) via its corresponding R package [50]. The NCT employs a permutation procedure, by which the (mean) difference (of a test statistic) between two networks is repeatedly calculated for random re-arrangements of observed data in order to create a reference distribution

(under the null hypothesis), through which a p-value can be obtained (please see details of our procedure in the supporting information). We used the NCT (in particular, three two-tailed tests; 1000 permutations each) to examine differences in global strength and strength centrality across each pair of our networks (i.e., left v. right; left v. centre; right v. centre). As a sensitivity check, differences in local and global expected influence, which take into account the sign of edge-weights, were also examined.

**Community (or factor) structure.** The community structure of each network was revealed with the use of two community-detection algorithms, the walktrap and spinglass algorithms, with the R packages 'EGA' [51] and 'igraph' [52], respectively. These modularity-based algorithms have been extensively used to reveal communities (which are mathematically equivalent to reflective latent variables) in networks comprising psychological measures [51]. A bootstrapped Exploratory Graph Analysis was further applied to validate the emerging factor structures (please see robustness section).

**Node predictability.** Finally, node predictability (which is analogous to $R^2$ in regression, as it quantifies the extent to which each belief is "explained" by the other beliefs in the network) was estimated through the R package 'mgm' [53] and was further visualized as a pie around each node in the networks.

**Robustness checks.** To scrutinize the robustness of any interesting patterns, a number of *post-hoc* robustness checks were conducted via the R packages 'bootnet' and 'EGAnet' [45,54]. First, since the NCT is sensitive to (extreme) sample imbalances between the comparison groups, random sampling was employed to extract a random yet equal number of participants per comparison group; the network comparisons were repeated on these groups. Second, a case-dropping subset bootstrapped procedure was employed to assess the stability of the centrality indices [45]. Third, the accuracy of the edge-weights was evaluated via a non-parametric bootstrapping procedure. Fourth, to validate the community structure of our networks, bootstrapped exploratory graph analysis (boot-EGA) was employed [55]. Finally, given concerns regarding the replicability of network structures [56, 57], we re-estimated our networks (using various estimation procedures) with an alternative definition of the centrist group and report these results (which pointed to equivalence) in our online supporting information.

## Results

### General comparisons

Table 1 gives descriptive statistics for the 18 political attitudes, as well as their differences across the ideological groups. From this table, we can see that those on the left and centre differed significantly across nearly all political beliefs. Conversely, fewer differences amongst policy preferences were identified between the right and the centre; however, there were still notable (median) differences for the following issues: P1 (Death Penalty); P2 (Spending money on Army); P7 (Higher Benefits for the Poor); P10 (Lower Taxes); P13 (Traditional Family Values); P14 (Monogamy); P16 (Brexit) (all p values < .001). The left and right differed significantly across all political beliefs, further indicating that the participants' self-assessment of their ideological orientation mapped onto common definitions of constraint in our data. As a further validation of the left-right self-placement scale we created a sum score for conservativism from the 18 individual policy items (reverse scoring where appropriate so that high scores always indicated right-wing policy preferences); as expected, the correlation with self-placement scale scores was highly significant, r = .504, p < .001.

### Comparison of political networks

The three political belief networks (Left, Centre, and Right) are displayed in Fig 1A. Stability and accuracy analyses (reported in the supporting materials) suggested that our network parameters were reliably estimated. Moreover, a complete re-analysis using random and equally-sized sub-samples and alternative estimation procedures replicated the current patterns, adding to their robustness.

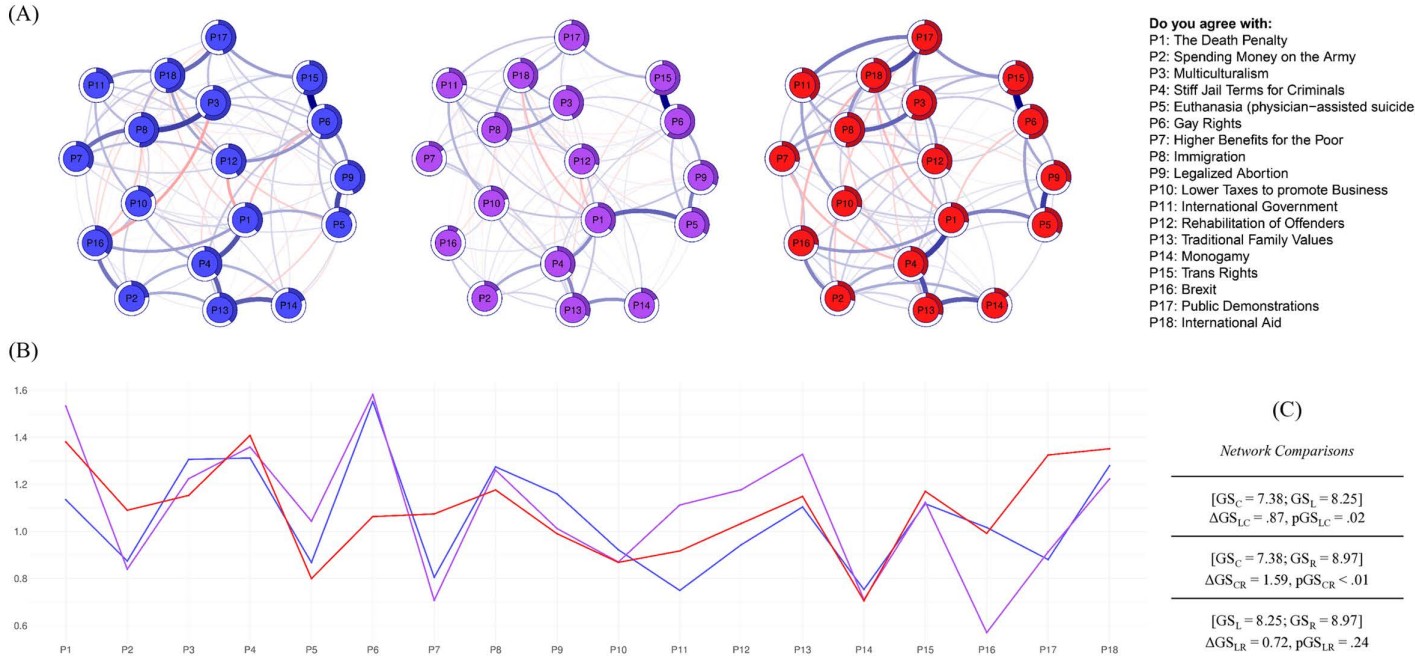

(A)

Do you agree with:
P1: The Death Penalty
P2: Spending Money on the Army
P3: Multiculturalism
P4: Stiff Jail Terms for Criminals
P5: Euthanasia (physician–assisted suicide)
P6: Gay Rights
P7: Higher Benefits for the Poor
P8: Immigration
P9: Legalized Abortion
P10: Lower Taxes to promote Business
P11: International Government
P12: Rehabilitation of Offenders
P13: Traditional Family Values
P14: Monogamy
P15: Trans Rights
P16: Brexit
P17: Public Demonstrations
P18: International Aid

(B)

(C)

*Network Comparisons*

$[GS_C = 7.38; GS_L = 8.25]$
$\Delta GS_{LC} = .87, pGS_{LC} = .02$

$[GS_C = 7.38; GS_R = 8.97]$
$\Delta GS_{CR} = 1.59, pGS_{CR} < .01$

$[GS_L = 8.25; GS_R = 8.97]$
$\Delta GS_{LR} = 0.72, pGS_{LR} = .24$

**Fig 1. Gaussian network models of political beliefs.** Section (A) shows models comprising political beliefs for each ideological group: Left, Centre, Right. Blue vs. red lines (or edges) represent positive vs. negative partial correlations, respectively, with thicker lines indicating stronger correlations. The predictability (variance explained) in each node is displayed around its circumference. Section (B) shows line graph displaying the strength centrality (i.e., extent of absolute connectivity level) for each node (variable) in the three networks. Section (C) shows Global Strength comparison results, revealing significantly higher global strength estimates for the Left and Right networks, compared to the Centrist network.

**Global strength.** Recall that a network's global strength reflects its absolute level of connectivity, which in our case was predicted to be increased at the ends of the political spectrum relative to the middle. The network comparison results supported this hypothesis by revealing significantly increased Global Strength estimates for the networks of those on the Left ($GS_L = 8.25$; $\Delta GS_{LC} = .87$, $pGS_{LC} = .02$) and Right ($GS_R = 8.97$; $\Delta GS_{CR} = 1.59$, $pGS_{CR} < .01$), compared to those in the Centre ($GS_C = 7.38$). Moreover, the networks of those on the Left and Right did not significantly differ in terms of the Global Strength ($\Delta GS_{LR} = 0.73$, $pGS_{CR} = .24$). These patterns were replicated in three sensitivity analyses: (1) using random and equally sized sub-samples of our data, (2) using raw rather than absolute sum of edge-weights, and (3) using a narrower definition of the centrist group (scores of 5 and 6 on the left-right orientation scale with a corresponding broadening of the other groups). As this finding was robust to these different checks, it suggests that the political belief systems of those individuals who identify themselves at the ends of the political spectrum are more strongly interconnected than those who identify in the centre (Fig 1C).

**Community structure.** The community structure of each network was examined using the walktrap and spinglass algorithms. These procedures revealed the exact same community structure in all three networks (see Fig 2). The first community reflected political attitudes that correspond to classic conceptions of *right-wing authoritarianism* [58,59]. The second community reflected a willingness to cooperate with, help and accept others even if different (e.g., P3, "Multiculturalism"; P8 "Immigration"); while it was more difficult to find a succinct name for this community we suggest that it reflects *altruism and cooperativeness*. The third and final community also reflected personal freedom and included issues that commonly align with social liberalism (e.g., P6, "Gay rights"; P15, "Trans rights"; P9, "Legalized abortion"); it was thus labelled as *personal liberalism*. Although our initial analyses pointed toward a two-factor structure (with the last two communities collapsing into a single liberal domain), the current three-factor solution was deemed more appropriate

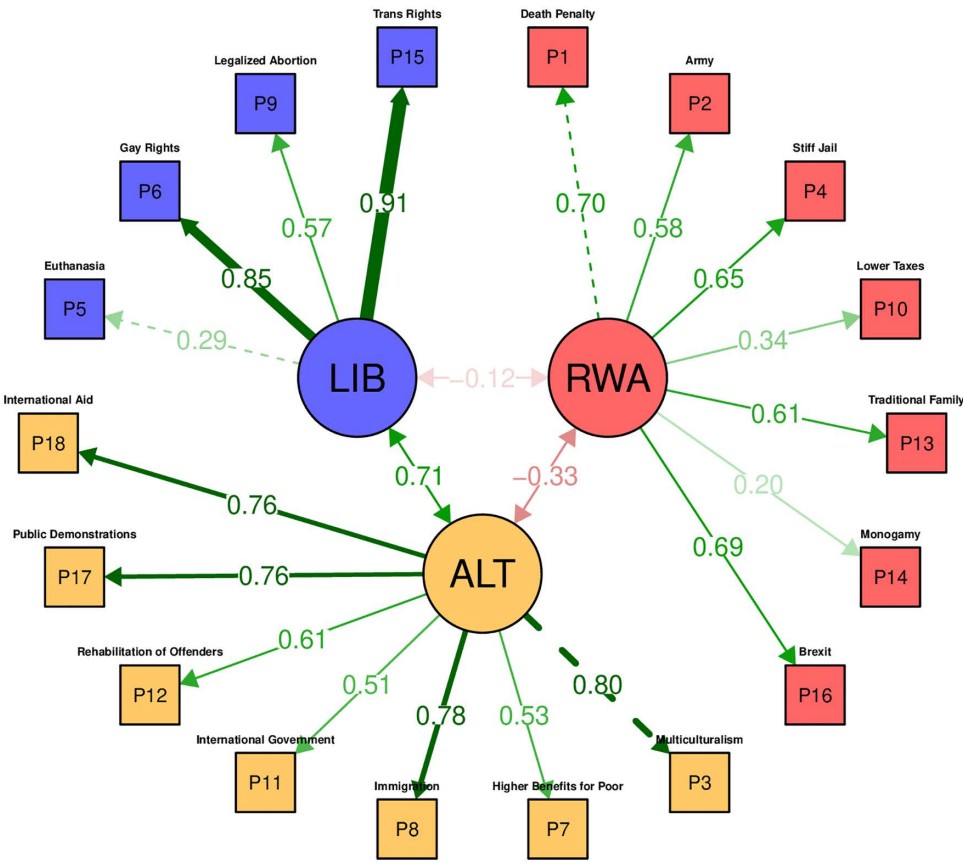

**Fig 2. Community structure of the entire sample.** Three communities are found in all political networks: namely, RWA = Right-wing authoritarianism, ALT = Altruism and cooperativeness, LIB = Personal liberalism.

given its high prevalence in a supplementary bootstrapping procedure (1,000 iterations; 72.5% frequency in bootstrapped distribution of replica networks) (see supporting information).

Factor scores on these three communities (which reflect the position of an individual on each latent variable) revealed that all three distinguished individuals from the different ideological groups in the expected way: For example, people on the left scored significantly higher on both altruism and cooperativeness and personal liberalism (compared to those on the centre and right); while those on the right scored significantly higher on right-wing authoritarianism (also, compared to the centre and left; see Table 2). A structural equation model (RMSEA = .07, CFI = 0.91,

**Table 2. Political group differences in community (factor) scores.**

| Community (Factor) | Mean Scores | | | Test Statistic, *p*-value | | |
|---|---|---|---|---|---|---|
| | Left | Center | Right | Center – Left | Center – Right | Left – Right |
| Right-Wing Authoritarianism | 3.53 (0.50) | 3.62 (0.48) | 3.74 (0.57) | 88987.5, < .001* | 86895.5, < .001* | 98400.5, < .001* |
| Altruism – Cooperativeness | 3.68 (0.49) | 3.58 (0.56) | 3.45 (0.46) | 84026, < .001* | 78139, < .001* | 102633.5, < .001* |
| Personal Liberalism | 3.59 (0.45) | 3.54 (0.47) | 3.48 (0.57) | 79260.5, < .001* | 75838, < .001* | 100608.5, < .001* |

Mean factor scores presented in each cell; standard deviation displayed in parentheses. Test Statistic (W) from the Wilcoxon rank sum test; * indicates that test remained statistically significant after a Bonferroni correction.

TLI = 0.90) showed that, as expected, child-rearing practices were a robust predictor of right-wing authoritarianism (β = 0.44, p < .001); and a strong negative predictor of altruism and cooperativeness (β = −0.32, p < .001) and personal liberalism (β = −0.31, p < .001).

**Node predictability.** The amount of explained variation in each node is outlined in Table 3 and is also graphically represented by the charts around nodes in Fig 1A. Several similarities and differences were detected in the predictability indices among the networks for those on the Left, Centre, and Right. For instance, P6 ("Gay Rights") and P15 ("Trans Rights") scored similarly high across the three networks. The predictability indices that differentiated the networks between the Left and Right included P7 ("Higher Benefits for the Poor"), P9 ("Legalized Abortion"), and P16 ("Brexit"), which were higher in the left network; and P5 ("Euthanasia"), P10 ("Lower Taxes"), and P11 ("International Government"), which were higher in the right network. Notably, the nodes that exhibited the highest levels of predictability within the Centre network scored similarly high in the networks for those on the Left and Right; these include: P3 ("Multiculturalism"), P4 ("Stiff Jail Terms"), and P14 ("Monogamy").

**Node centrality.** The strength centrality indices for each node (i.e., the absolute sum of all edges linked to a given node) in all networks are graphically displayed in Fig 1B. From this figure, it can be observed that a particular node scores the highest for the Left and Centre networks, namely, P6 ("Gay Rights"). By contrast, node P17 ("Public Demonstrations") scored the highest in the Right network. The NCT validated the central position of these nodes in their respective networks but also revealed several other significant differences in centrality indices across the political belief networks of the Left, Centre, and Right. Notably, node P16 ("Brexit") scored significantly lower in strength centrality in the centrist network, compared to the left and right ones; node 17 ("Public Demonstrations") scored significantly higher in strength centrality in the Right network, compared to the Left and Centre ones.

**Table 3. Predictability indices of particular nodes across the three political belief networks.**

| Variable | Predictability Indices | | |
|---|---|---|---|
| | Left | Centre | Right |
| P1 (Death Penalty) | 0.372 | 0.357 | 0.272 |
| P2 (Spending money on Army) | 0.239 | 0.161 | 0.271 |
| P3 (Multiculturalism) | 0.589 | 0.432 | 0.580 |
| P4 (Stiff Jail Terms) | 0.369 | 0.343 | 0.363 |
| P5 (Euthanasia) | 0.129 | 0.342 | **0.336** |
| P6 (Gay Rights) | 0.540 | 0.602 | 0.514 |
| P7 (Higher Benefits for the Poor) | **0.347** | 0.159 | 0.279 |
| P8 (Immigration) | 0.518 | 0.361 | 0.516 |
| P9 (Legalized Abortion) | **0.441** | 0.323 | 0.293 |
| P10 (Lower Taxes) | 0.173 | 0.205 | **0.285** |
| P11 (International Government) | 0.227 | 0.246 | **0.409** |
| P12 (Rehabilitation of Offenders) | 0.408 | 0.283 | 0.339 |
| P13 (Traditional Family Values) | 0.379 | 0.305 | 0.294 |
| P14 (Monogamy) | 0.196 | 0.182 | 0.149 |
| P15 (Trans Rights) | 0.563 | 0.589 | 0.592 |
| P16 (Brexit) | **0.362** | 0.101 | 0.264 |
| P17 (Public Demonstrations) | 0.450 | 0.343 | 0.557 |
| P18 (International Aid) | 0.517 | 0.397 | 0.550 |

Predictability indices in bold represent the ones that are highest uniquely for each network.

## Discussion

In this study, we sought to evaluate how the structure of political belief systems varied as a function of political ideology. To this end, we constructed three Gaussian network models of political beliefs for people who classified themselves as being on the left, centre, and right of the left-right ideological spectrum.

This work extends previous research that has applied network models to analyse political beliefs, for example by Dalege and colleagues [32–34], Boutyline and colleagues [35] and Brandt and colleagues [36]. However, the nodes in our networks were restricted to attitudes towards a range of policies. In contrast to previous analyses, we deliberately chose not to include behavioural measures such as voting history [33] or symbolic demonstrations of political affiliation such as party membership [35] because past research has shown that attitudes towards such constructs are strongly influenced by non-ideological factors [25]. More importantly, had we included these variables in our networks, we would have undermined our goal of establishing whether ideological positioning (which we operationalised independently of specific beliefs by dividing people into left, centre or right groups based on self-allocation on the political spectrum) effected the network structure. With regard to the validity of the self-allocation method, we note that it has been common practice in election surveys [43] and research in political psychology [3].

Our primary hypothesis, derived from Converse's analysis of the belief systems of mass publics [21] was that those on the left and right would have more interconnected networks compared to those in the centre. Our first contribution to the literature on political ideologies is to show that this hypothesis is supported by a permutation test on the Global Strength (absolute sum of edge-weights) of the three networks. In particular, the political belief networks of those on the left and right exhibited significantly increased Global Strength estimates, compared to the political network of those in the centre. Of note is that we compared the three networks in terms of their absolute level of connectivity (although the same patterns emerged in a robustness check when taking the raw values of edge-weights into consideration, shown in the supporting information), to explicitly examine Converse's account [21]. Overall, these results show that those who identify at the political extremes are *similar in the structure of their political beliefs* even though they are *different in their actual beliefs* (those on the left disagreed with those on the right about all of the eighteen policies).

Importantly, this finding is consistent with recent studies showing that those who self-position at the extremes of the left-right spectrum are more dogmatic and hold their beliefs more consistently over time and with greater certainty compared to those in the centre [26,27]. These are properties that are predicted by Converse's account because both certainty and resistance to attitude change should be determined by the strength of the connections between nodes (highly interconnected nodes being unable to change without adjustments to other nodes). They are also consistent with previous network structures of attitudes, which have shown that attitude strength as measured by future behaviour is predicted by the strength of connections within an attitude network [34].

Despite their differences in global connectivity, the three political networks exhibited similar relational structures, as evidenced by their convergence in a final factor structure that comprised three communities. Importantly, our second contribution to the literature is that our data-driven communities *did not* match the division between social and economic conservatism found in factor-analytic investigations by other scholars [9,10]. Nonetheless, and in contrast to previous conclusions about the electorate, the communities were interpretable as political constructs aligned with classic accounts of right-wing authoritarianism [58,59], altruism and cooperativeness towards others, and personal liberalism (i.e., attitudes about the freedom of the individual to choose their life course). Two points about this observation are perhaps worth noting. First, the convergence into the same factor (or community) structure in the three networks does not, of course, imply similar attitudes toward these constructs; when scores on the individual items were compared, for example, those on the left and right differed in the expected direction on the items in the right-wing authoritarianism community (associated with the right), as well as items belonging to the remaining two communities (associated with the left, though attitudes toward euthanasia were an exception). Second, the expected relationship between scores on the three communities and a widely used nonpolitical measure of authoritarian personality traits, attitudes towards child rearing [37,38], suggests that the communities have some validity.

Local features of the networks are more difficult to interpret because they relate to each other in complex ways. In general, variation in the centrality statistics (which measured the extent to which each political attitude was connected to others within the network) approximately matched variation in node predictability (the extent to which each belief could be predicted given knowledge of all other beliefs). Our third contribution is that the highly interconnected node of 'Gay rights' has a large predictability index in all three networks and is very central in the right and centre networks (although less so for the left). Scholars have previously shown less acceptance of sexual minorities by people on the right [60], as well as in those scoring high on right-wing authoritarianism and social dominance orientation [61]. The present findings raise the interesting possibility that attitudes toward sexual minorities may play an important and yet under-recognised role in determining an individual's choice of ideology. An unmeasured factor in this study that might be important in this respect is disgust sensitivity, which has been found to be a modest predictor of conservative and authoritarian attitudes [62] and a stronger predictor of negative attitudes toward gay men [63]. In general, this finding is consistent with previous research showing that social conservatism, in contrast to economic conservatism, is strongly linked to personality factors [64,65] and to evolved implicit biases that influence values and attitudes [66].

The high centrality shown for "public demonstrations such as 'taking the knee' to acknowledge discrimination against ethnic minorities" is more difficult to account for in terms as fundamental evolutionary biases, as these kinds of protests were prominent in the UK media at the time that the data were collected but have since received much less attention. Scores on this item are perhaps best interpreted as markers of other attitudes, for example ethnocentrism, or resistance to challenges to the status quo.

In the predictability indices, it is possible to, again, see similarities between the left and right compared to the centre. For four of the constructs – namely, "Multiculturalism," "Higher Benefits for the Poor," "International Aid" and, albeit slightly less clearly, "Public Demonstrations"– predictability indices are higher at both ends of the ideological spectrum than in the centre. The centrality of "Brexit" was significantly lower in the centrist network than the left- and right-wing ones (for which scores where similarly high), and this pattern also emerged in the predictability indices themselves. The fact that this node falls into the community we have characterised as "right-wing authoritarianism" suggests that support for Brexit in the UK at the time of the survey may have been more connected to authoritarian attitudes than political ideology *per se*.

## Limitations

The current study should be viewed in light of certain limitations. Although we used a large, diverse sample that was fairly representative of the UK population, some of the political constructs included (e.g., Brexit) were unique to the UK political landscape. There were more items that related to social conservatism than to economic conservatism, which may also limit the generalizability of our findings. It is possible, for example, that a different structure would have emerged if we had included more economic items, perhaps with a specific community relating to economic beliefs, and perhaps reducing the influence of social items, for example relating to gay and trans rights. Given recent observations of cross-cultural differences in the relationship between social and economic conservatism [18] it is also possible that different communities of beliefs would be found in different regions of the world. Similarly, the survey was conducted during the COVID-19 pandemic at a time of great social turbulence, which may also limit generalizability to different periods of time. Although we doubt whether cross-cultural differences would be found with respect to our primary observation that those who locate themselves at the political extremes have more closely connected networks of beliefs (because we think this is a general property of human belief systems) it would certainly be desirable to replicate this work in other geographical locations.

It is also worth noting that the procedure we employed here to categorise individuals into the left, centre, and right is more data-driven than theoretical. However, as already noted, this is standard practice in political surveys and research in political psychology and there is good evidence that most people can assign themselves along the left-right political

spectrum [11] and describe the distinction between the two poles in similar terms [12]. Moreover, our between-group comparisons show that these self-categorisations reflect actual differences in policy preferences, and there was the expected correlation between overall support for right-wing policies and self-placement on the left-right scale. Finally, our sensitivity analyses show that our main findings were unaffected when we narrowed the definition of the centre group and widened the definitions of the left and right groups.

We must also acknowledge that the childrearing items we used to validate our community structure had marginal reliability. Our structural equation model found robust relationships with each of the communities. As we would expect the poor reliability of these items to attenuate these relationships, we are reassured by their strength and by the fact that the model fit indices were satisfactory. Nonetheless, this finding should be treated with caution and future studies might consider a wider range of measures to validate observed community structures of political beliefs.

Finally, much network psychometric research has employed the Ising model and has included beliefs along with other variables of similar nature (e.g., feelings). However, in this study, we have modelled political beliefs in a Gaussian manner because Ising models only accommodate binary data. Albeit not a limitation in and of itself, it is worth noting that, in practice, the two models often produce similar results [67].

## Conclusions, implications and further directions

Converse [21] distinguished voters not only in terms of their ideology but also according to whether they were 'high information' vs. 'low information' voters. In the relatively few network studies of political beliefs published prior to this one, Boutyline [35] and Dalage [34] reported that the former had more strongly connected networks than the latter. In as much as political ideology is likely to drive the seeking of political information and knowledge, the two concepts – ideology and information – are likely to be closely related. Nonetheless, it will be useful in future studies to try and disentangle these effects, especially as, in all likelihood, some individuals in the centre have carefully thought out but idiosyncratic belief systems whereas others are simply uninterested in politics.

Network models might also be used to examine the influence of more fundamental psychological processes on belief systems. For example, we earlier mentioned the possible role of disgust sensitivity in political beliefs [62,63]. An implication of our network account of political attitudes is that individuals high in disgust sensitivity should show more interconnected networks, with greater centrality and predictability indices for attitudes toward sexual minorities, compared to people low on disgust sensitivity.

Finally, we would like to draw attention to the fact that our research has been driven by an interest in belief systems in general, rather than political beliefs specifically. Inspired by Converse's analysis (which he was clear was not limited to political ideologies) we expect that highly interconnected networks will be a feature of all socially consequential beliefs systems that are held with great certainty, and which are stable over time. Interestingly, excessive certainty and resistance to change have been identified as important properties of the delusional beliefs of psychiatric patients [68]. In previous research, we have shown that the paranoid delusions of psychiatric patients differ from the subclinical forms of paranoia observed in the healthy population, because they consist of a more highly interconnected network of beliefs about the trustworthiness of others and likelihood of threats [69]. In this respect at least, the pathological beliefs of psychiatric patients and extreme ideologies may have something in common [70]. It would be useful in future research to examine the same network properties in relationship to other kinds of belief systems, for example religious beliefs and conspiracy theories.

## Supporting information

**S1 File.  Supporting information.** This file contains a detailed methodological rationale, sensitivity analyses, robustness analyses, network replicability, and centrality metrics.
(PDF)

## Author contributions

**Conceptualization:** Richard P. Bentall, Orestis Zavlis, Philip Hyland, Orla McBride, Kate Bennett, Todd K. Hartman.

**Data curation:** Richard P. Bentall.

**Formal analysis:** Orestis Zavlis, Orla McBride, Todd K. Hartman.

**Investigation:** Richard P. Bentall.

**Methodology:** Richard P. Bentall, Orestis Zavlis.

**Project administration:** Richard P. Bentall.

**Writing – original draft:** Richard P. Bentall, Orestis Zavlis, Philip Hyland, Orla McBride, Kate Bennett, Todd K. Hartman.

**Writing – review & editing:** Richard P. Bentall, Philip Hyland, Orla McBride, Kate Bennett, Todd K. Hartman.

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
