## [Decision Letter · Decision Letter 0]

26 Feb 2025

Dear Dr. Bentall,

**Both reviewers agree that your manuscript has the potential to make a valuable contribution to the existing literature. However, they both raise concerns regarding the empirical and theoretical aspects of your work. In line with their recommendations, I also suggest that you thoroughly review the network literature in political science, as it may both inform and extend your arguments in the manuscript.  **plosone@plos.org . A rebuttal letter that responds to each point raised by the academic editor and reviewer(s). You should upload this letter as a separate file labeled 'Response to Reviewers'.A marked-up copy of your manuscript that highlights changes made to the original version. You should upload this as a separate file labeled 'Revised Manuscript with Track Changes'.An unmarked version of your revised paper without tracked changes. You should upload this as a separate file labeled 'Manuscript'.

We look forward to receiving your revised manuscript.

Kind regards,

Cengiz Erisen

Academic Editor

PLOS ONE

**Journal Requirements:**

1. When submitting your revision, we need you to address these additional requirements. Please ensure that your manuscript meets PLOS ONE's style requirements, including those for file naming. The PLOS ONE style templates can be found at https://journals.plos.org/plosone/s/file?id=wjVg/PLOSOne_formatting_sample_main_body.pdf and https://journals.plos.org/plosone/s/file?id=ba62/PLOSOne_formatting_sample_title_authors_affiliations.pdf 2. Thank you for stating the following financial disclosure: The data collection for this study was funded by an ESRC grant to Bentall (CI) and Hartman (co-I) from the Economic and Social Research Council, A longitudinal mixed-methods population study of the UK during the COVID-19 pandemic: Psychological and social adjustment to a global threat,  ES/V004379/1    Please state what role the funders took in the study.  If the funders had no role, please state: "The funders had no role in study design, data collection and analysis, decision to publish, or preparation of the manuscript." If this statement is not correct you must amend it as needed. Please include this amended Role of Funder statement in your cover letter; we will change the online submission form on your behalf. 3. Thank you for stating the following in the Acknowledgments Section of your manuscript: The initial stages of the Covid 19 Psychological Research Consortium project were supported by start-up funds from the University of Sheffield (Department of Psychology, the Sheffield Methods Institute and the Higher Education Innovation Fund via an Impact Acceleration grant administered by the university) and by the Faculty of Life and Health Sciences at Ulster University. The research was subsequently supported by the UK Economic and Social Research Council under grant number ES/V004379/1 and awarded to RPB, TKH, OMcB, and KB and others. The present analysis was further supported by a grant from Higher Education Innovation Fund awarded by the University of Manchester. The funders had no role in study design, data collection and analysis, decision to publish or preparation of the manuscript. We note that you have provided funding information that is not currently declared in your Funding Statement. However, funding information should not appear in the Acknowledgments section or other areas of your manuscript. We will only publish funding information present in the Funding Statement section of the online submission form. Please remove any funding-related text from the manuscript and let us know how you would like to update your Funding Statement. Currently, your Funding Statement reads as follows:  The data collection for this study was funded by an ESRC grant to Bentall (CI) and Hartman (co-I) from the Economic and Social Research Council, A longitudinal mixed-methods population study of the UK during the COVID-19 pandemic: Psychological and social adjustment to a global threat,  ES/V004379/1 Please include your amended statements within your cover letter; we will change the online submission form on your behalf.  

Reviewers' comments:

Reviewer's Responses to Questions

**Comments to the Author**

1. Is the manuscript technically sound, and do the data support the conclusions?

Reviewer #1: Yes

Reviewer #2: Yes

2. Has the statistical analysis been performed appropriately and rigorously?

Reviewer #1: Yes

Reviewer #2: Yes

3. Have the authors made all data underlying the findings in their manuscript fully available?

Reviewer #1: Yes

Reviewer #2: Yes

4. Is the manuscript presented in an intelligible fashion and written in standard English?

Reviewer #1: Yes

Reviewer #2: Yes

**Reviewer #1:**  This is an interesting manuscript that uses network analysis to look at old ideas of opinion constraint (a la Converse) using a) new data and b) set in the British public. I like the effort, and appreciate the analysis. That said, a few comments that could clarify the approach and strengthen the contribution:

1) The authors should cite additional, empirical work that has been done on network approaches to political ideology. Particularly relevant is an AJS paper by Baldassarri and Goldberg (2014) that uses ANES data and relational class analysis (RCA). The aforementioned paper does a better job than the present paper of situating efforts to look at ideologies as networks in existing literatures (it also has a temporal component that the present paper does not, though the authors are upfront about this limitation).

2) The authors should explain the methods in the paper for the non-network researcher. The authors mention that nodes are issues, but readers not familiar with network analytic frameworks will have a hard time understanding the data structure and how the edges are estimates. Just a bit of elaboration would likely help/solve this.

3) For the network reader, the choice of this network modeling approach vs. others deserves a mention.

4) The one robustness check I’d like to see/I think would be useful: the authors divide the public based on a 10-point self-identified ideology measure, and then estimate 3 networks. What do the results look like when what counts as a middle category (moderates) is changed to just 5,6? What happens when the left and right are broadened/narrowed? In other words, are the differences in structure there if the authors play with how they split the data initially?

**Reviewer #2:**  I found this manuscript to be very well-written and the research to be interesting. I see much value in the network-analytic approach even though I can’t say that I am well-versed in it. Thus, one caveat in my review is that I am unable to say anything about the nuances of those analyses. Below are my detailed comments:

1. While I understand that the three communities emerged in data-driven fashion and I am actually fond of such data-driven efforts, I found the distinction between interpersonal and intrapersonal liberalism to be problematic because, first, the terms interpersonal and intrapersonal do not closely match “society-focused” and “personal”, respectively. The former set is confusing because of how these terms are used in social and personality psychology. The latter set of terms (used parenthetically by the authors) are much clearer. Second, examining the issues that make up these communities, I feel that even the distinction between society-focused and personal does not apply very well. In other words, I fail to see how one term but not the other applies to something like P7 or P6. From what I can see the common thread in the “intrapersonal” cluster is much clearer: all issues involve the right to make fundamental personal choices about one's body, identity, or life. The “interpersonal” cluster has a broader scope but consistently emphasizes international cooperation and social support rather than isolation or punishment. In addition, the “woke” concept does not help me here (partially because I do not closely follow current movements in Western society) and its relevance is supported by a journalistic piece. I would like to see something in the scientific literature instead or in addition. The same point holds for whatever terms are used to mark these two communities: Ideally, they should be discussed in the light of earlier findings. Do these just emerge as totally surprising new clusters or are they similar to any earlier conceptual (or data-driven) offerings?

2. Perhaps the most interesting finding is that the data-driven approach yields three clusters of attitudes that do not map onto social and economic conservatism. But I wonder if this might have anything to do with the list of issues used in data collection. The manuscript does not inform us who else used this or a similar list in their research (apart from Wilson in 1968) and say anything about reliability and validity. Only four of the issues seem to be more about economic than social conservatism (P2, P7, P10, P16—the latter assuming Brexit is primarily about economic sovereignty/regulation). I think 9 of the issues are clearly about social conservatism and the remaining 5 are a mix. Would the authors agree with this and if so, is this imbalance of issue representation relevant to the divergence of their findings from the literature? In other words, how confident should we be that the key economic issues concerning the UK are represented? A related note is that the authors claim on p. 5 that they “anticipated” this divergence but do not explain how they were able to.

3. More information on the child-rearing preferences measure could be added, as well. Was this identical to one of the references used to support its inclusion? Is there any reliability and validity information? This is called “child-rearing practices” on p. 22, which should be corrected for consistency and accuracy.

4. Some of the findings reported here, that the authors usually discuss together with references #26 and #27 and Converse’s approach, might just be about attitude strength on which there is a sizeable literature. Situating this research more broadly in that literature, even if briefly (e.g., with a footnote), would enrich this manuscript. This is consistent with the authors’ view that Converse’s model and their work is not narrowly about political attitudes. However, I fear that bringing the attitude strength literature into focus raises the question “what is new in the current findings concerning interconnectedness of attitudes?” Would the attitude strength literature independently (of Converse’s model) predict the higher interconnectedness of attitudes on the extreme ends of the political spectrum? An in-depth discussion of the attitude strength literature would be distracting but some more detail on this would enable the readers to place the current work in broader context and to appreciate the novelty or convergence to earlier findings.

5. I think the manuscript does not make it clear how and why Converse’s approach (“beliefs as a network”) “contrast sharply” (p. 8) with factor-analytic approaches. What are the different assumptions of each? How do the current findings uniquely support the former approach as opposed to the latter? Why exactly are factor-analytic approaches “ill-suited for testing” (p. 10) Converse’s model? Whereas the authors mention that “the network approach rejects the idea that beliefs ... necessarily cluster together into recognizable ideologies ... because of a common underlying cause such as specific cognitive traits” (p. 10), they later go on to identify three communities and also suggest the possibility that “hidden psychological processes” (e.g., disgust sensitivity; authoritarian traits) may be related to this organization. How do the latter features of the authors’ work differ from how they think factor-analytic and network approaches differ (e.g., communities versus clusters; common underlying causes versus hidden psychological processes)? More detailed explanation on these would benefit readers.

6. The authors mention in the abstract that the sample is “representative ... of UK adults.” In the Discussion, they state that the sample is “fairly representative” (p. 28). In the Methods, we learn that this sample is a wave that is among a set of waves that is considered to be “slightly less representative” than an initial wave. Instead of these statements, perhaps we could be told how representative the sample is with regard to important aspects of the UK adult population more concretely.

7. The abstract reports sample size as 1,634. On p. 13, we learn that it is 1,643 + 415. Is something wrong in either section?

8. I did not see anything about patterns of missing data in the manuscript of the supplementary information document. It could be checked and reported.

9. Since the test statistic from Wilcoxon rank sum test does not indicate effect size, a separate effect size indicator could be provided for Table 1.

10. About the figures:

11. I have many notes about the figures:

a. Nearly every aspect of the network visualizations is unfamiliar to the majority of readers in Psychology (I’m not sure about other fields but probably the same). So a comprehensive note underneath these figures seems like a good idea to me. At the least, in Figure 1, the vertical axis on the top panel could be labeled either on the figure or in a note. I assume that this axis is edge-weights. I’m also not sure what the lines in the figure (as opposed to circles) represent; maybe confidence intervals of estimated edge-weights, though I’m not sure why those would not be symmetrical. I assume the centre and left circles overlap completely in P1. Perhaps they could be jittered. For the bottom panel, I think the “pie around each node” represents node predictability. This is explained in the text but is disconnected from the figure. It’s better to place this in a note below the figure.

b. In Figure 2, it seems that the top left network represents leftists, the top right one represents centrists, and the bottom network represents rightists. Labeling these networks would be better. What is the difference from the bottom panel of Figure 1 apart from the additional representation of community structure. These two sets of 3 networks seem redundant. If this is correct, I would remove the bottom panel of Figure 1.

c. The panels in Figure 1 are called Figure 1a (I assume the top panel) and 1b (the bottom panel) in the text but not labeled in the figure. A large “A” or “B” next to these panels and/or a note should be added.

d. The authors mention that Figure 1b shows “that a particular node scores the highest for the Left and Centre networks, namely, P6 (“Gay Rights”). By contrast, node P17 (“Public Demonstrations”) scored the highest in the Right network.” I assumed that figure 1b was the bottom panel but looking at this figure, what I see is that line thickness varies significantly between connections and represents connection strength; some thin connections might have very low weights while thicker ones have much higher weights; and P6 appears to have several relatively thick connections especially in Leftists. However, it's not immediately apparent that P6's connection weights would sum to a higher total than nodes like P3 or P12, which appear to have both numerous and thick connections. Without the actual edge weight data, we can't verify their claim about P6 having the highest strength centrality. If Figure 1b is the top panel, then once again, I cannot see how P17 is the most central node for rightists. There are rightist nodes higher on the Y-axis and circle size does not help here. I did not see any data in the Supplementary Information that could help with this. Perhaps I’m missing something due to not being so familiar with network analyses but whatever I’m missing could also be missed by other readers. So, further explanation would be helpful.

12. In Table 2, we are presented with “predictability indices ... that are highest uniquely for each network.” I understand the general approach but do not know if there is a convention in network analysis for deciding on these or whether the authors had their own specific criteria. This could be clarified.

Minor points:

- “conservativism” in the abstract should be corrected as “conservatism”

In sum, I think the current manuscript could be published with revision in presentation (i.e., edits to text and figures). I congratulate the authors on their work.

**Do you want your identity to be public for this peer review?** For information about this choice, including consent withdrawal, please see our Privacy Policy

Reviewer #1: No

Reviewer #2: **Yes: ** S. Adil Saribay

---

## [Author Response · Author response to Decision Letter 1]

26 Apr 2025

The authors would like to thank the two reviewers for their diligence in reviewing our manuscript and especially for the detailed and very helpful suggestions by Reviewer 2, who identifies himself as Dr S. Adil Saribay. We have tried to respond to the reviewers’ recommendations as much as possible, and we feel confident that, thanks to their advice, our paper is now much stronger.

Reviewer 1

1. This reviewer asks us to include the Baldassari and Goldberg (2014) paper in the background material included in our introduction

RESPONSE: We are grateful for the referee for drawing our attention to this study which is indeed relevant to our own work. The approach used in the study is highly complex and somewhat different from our own, but we have provided a brief summary of the main finding on page 9 of the revised manuscript.

2. We are asked to provide an expanded explanation of network methods that will be more comprehensible to people who are not familiar with the approach.

RESPONSE: We have expanded our explanation of network approaches on pages 10 and 11 of the manuscript in a way that will hopefully make the approach easier to grasp for someone who has no familiarity with these methods.

3. This reviewer argued that an additional robustness check would be valuable: “The authors divide the public based on a 10-point self-identified ideology measure, and then estimate 3 networks. What do the results look like when what counts as a middle category (moderates) is changed to just 5,6? What happens when the left and right are broadened/narrowed? In other words, are the differences in structure there if the authors play with how they split the data initially?”

RESPONSE: Many thanks for recommending this excellent robustness check. We had in fact previously explored a similar sensitivity check, namely whether our results held if we randomly selected different subpopulations from each of the three groups (i.e., random and equal samples for left-wing, centrist – defined as scoring 4 to 7 on the measure of ideology – and right-wing populations). We had reported these findings in our Supplement Section III and showed that they pointed to the exact same results: namely, that centrist populations exhibited weaker network connectivity compared to the left-wing and right-wing.

We have now repeated the same procedure for the centrist population defined by scores of 5 to 6, rather than 4 to 7, as suggested by this referee. We found that our main result (significantly higher network connectivity in the left-wing and right-wing populations versus the centrist populations) remains invariant in this case. We have added the results of this new analysis in our Supplement Section III and comment on them in our Results section on page 23 and in the discussion on page 32.

Reviewer 2

1. This reviewer says that the distinction between interpersonal and intrapersonal liberalism is problematic because the terms interpersonal and intrapersonal do not closely match “society-focused” and “personal”, respectively. He makes the interesting suggestion that the common thread in what we termed the “intrapersonal” cluster is that all of the issues involve the right to make fundamental personal choices about one's body, identity, or life, and that the elements in the “interpersonal” cluster have a broader scope but consistently emphasizes international cooperation and social support rather than isolation or punishment. The referee also thinks that our use of the term “woke” is not appropriate for an academic paper.

RESPONSE: While the labelling of the communities is to some extent a subjective issue, we are happy to relabel the two contested communities as “personal liberalism” and “altruism and cooperativeness” (page 23 of the results and pages 29 in the discussion). We also agree that it was a mistake to use the term ‘woke’, and we have therefore removed it from the manuscript.

2. The reviewer says: “Perhaps the most interesting finding is that the data-driven approach yields three clusters of attitudes that do not map onto social and economic conservatism. But I wonder if this might have anything to do with the list of issues used in data collection. The manuscript does not inform us who else used this or a similar list in their research (apart from Wilson in 1968) and say anything about reliability and validity. Only four of the issues seem to be more about economic than social conservatism (P2, P7, P10, P16—the latter assuming Brexit is primarily about economic sovereignty/regulation). I think 9 of the issues are clearly about social conservatism and the remaining 5 are a mix. Would the authors agree with this and if so, is this imbalance of issue representation relevant to the divergence of their findings from the literature?”

RESPONSE: The items were based on Wilson’s measure, which the authors adapted to include items that reflect issues that seem important at the present time.

The validity of the items is established by the fact that the three groups, Left, Centre and Right, differed on them exactly in the way expected as shown in Table 1, and we have now added a comment to this effect on page 32 of the discussion. The reliability of the measure is not relevant to our analysis because network psychometrics do not assume that the items belong to a scale (i.e., that they are indicators of an underlying latent trait), but nonetheless, it has been possible to calculate a reliability coefficient with the direction of each item determined by the group differences, and it is high, 0.83. We now note this on the scale description in the methods section, page 15.

We agree with this reviewer’s suggestion that the communities that emerged may have been affected by the balance between economic and social items. We now acknowledge this in the discussion on page 31.

3. The reviewer requests more detail about the childrearing scale and asks us to check our naming of it throughout.

RESPONSE: This scale is a very widely used measure in political science. We have expanded the description of its origins in the methods section, page 15, and also calculated its reliability which is marginal, although we do not think this is a critical weakness as the scale is only used in a supporting analysis as a way of validating our community structure. We acknowledge this low reliability in the limitations section on page 32.

4. The referee asks us to explain how our findings relate to the large literature on attitudes strength, perhaps in a footnote.

RESPONSE: The literature on attitude strength is, as the referee states, very large and therefore beyond the scope of our paper. However, in our submitted manuscript we discussed this implicitly by commenting on how our findings are consistent with research on the certainty and resistance to change of political attitudes. We have slightly expanded this paragraph on page 30 of the revised discussion section to make this point clearer and to link to Dalege’s work on causal attitude networks.

5. The referee asks us to make it clearer how and why Converse’s approach (“beliefs as a network”) “contrast sharply” with factor-analytic approaches.

RESPONSE: We think our rewritten account of network approaches, made in response to Referee #1’s point 2 goes some way to addressing this issue. However, we have removed the word ‘sharply’ from the first paragraph of our account of Converse.

6. The referee asks us to provide more information in the methods about how representative the sample is with regard to important aspects of the UK adult population more concretely.

RESPONSE: We report that our sample was stratified by age, sex and household income in quotas that represented the structure of the UK population according to the UK Office of National Statistics. We also report the proportion who were female, the proportion who came from each of the constituent administrations in the UK, the proportion who were graduates, the proportion who describe themselves as working class, and the sample’s voting behaviour in the Brexit referendum. In two instances (the proportion of graduates and voting) we report that our sample has drifted from the comparable population figures because of attrition across survey waves. We have added a reference to the latest Office of National Statistics population report (page 14) but, otherwise, we do not understand how we can make our account of our sample more concrete than it is.

7. The referee points out that he abstract reports sample size as 1,634 but on p. 13, we learn that it is 1,643 + 415.

RESPONSE; We thank the referee for pointing this out. The abstract was in error and has been corrected.

8. The reviewer asks us to report patterns of missing data.

RESPONSE: There was no missing data because our survey software required responses for every item before moving on. This is now mentioned at the beginning of the measures section.

9. This reviewer suggests that we include effect sizes in Table 1

RESPONSE: After some thought, we have decided not to include effect sizes because the figure is crowded as it is and because these are not relevant to the hypotheses that we have tested in our paper.

10. This reviewer has pointed to many issues with our figures, partly due to my accidental inclusion of the wrong versions of the figures in my last submission. He makes many important and useful observations and we are very grateful for his diligence and the opportunity to improve them:

“Nearly every aspect of the network visualizations is unfamiliar to the majority of readers in Psychology (I’m not sure about other fields but probably the same). So, a comprehensive note underneath these figures seems like a good idea to me.”

RESPONSE: We completely agree that there should be a comprehensive figure caption to explain what each figure illustrates. We have now updated all of our figures in light of the comments and included a detailed caption to more clearly explain them.

“At the least, in Figure 1, the vertical axis on the top panel could be labelled either on the figure or in a note. I assume that this axis is edge-weights. I’m also not sure what the lines in the figure (as opposed to circles) represent; maybe confidence intervals of estimated edge-weights, though I’m not sure why those would not be symmetrical. I assume the centre and left circles overlap completely in P1. Perhaps they could be jittered. For the bottom panel, I think the “pie around each node” represents node predictability. This is explained in the text but is disconnected from the figure. It’s better to place this in a note below the figure.”

RESPONSE: Again, thanks for these suggestions! We have now completely redrawn our figures to make it clearer to the readers that we are presenting three key findings: first and foremost, the three network structures for the left, centre, right; second, the centrality of each node within the network; finally, the network comparison tests. (We apologise that this was often unclear our previous submission, e.g., that the line graph shown at the top of our previous Figure 1, was meant to represent the centrality of each node; we no longer include this in the present submission). We also include a comprehensive caption underneath that states what each panel illustrates. We hope this major update addresses the reviewer’s point.

“In Figure 2, it seems that the top left network represents leftists, the top right one represents centrists, and the bottom network represents rightists. Labelling these networks would be better. What is the difference from the bottom panel of Figure 1 apart from the additional representation of community structure. These two sets of 3 networks seem redundant. If this is correct, I would remove the bottom panel of Figure 1.”

RESPONSE: We agree that it is redundant to display the same networks twice. We have thus excluded the networks from Figure 2 and simply outline the factor structure of our three networks.

“The panels in Figure 1 are called Figure 1a (I assume the top panel) and 1b (the bottom panel) in the text but not labelled in the figure. A large “A” or “B” next to these panels and/or a note should be added.”

RESPONSE: Many thanks for catching this omission from our end! We have now included large (A), (B), and (C) letters to clearly outline the three panels of Figure 1.

“The authors mention that Figure 1b shows “that a particular node scores the highest for the Left and Centre networks, namely, P6 (“Gay Rights”). By contrast, node P17 (“Public Demonstrations”) scored the highest in the Right network.” I assumed that figure 1b was the bottom panel but looking at this figure, what I see is that line thickness varies significantly between connections and represents connection strength; some thin connections might have very low weights while thicker ones have much higher weights; and P6 appears to have several relatively thick connections especially in Leftists. However, it's not immediately apparent that P6's connection weights would sum to a higher total than nodes like P3 or P12, which appear to have both numerous and thick connections. Without the actual edge weight data, we can't verify their claim about P6 having the highest strength centrality. If Figure 1b is the top panel, then once again, I cannot see how P17 is the most central node for rightists. There are rightist nodes higher on the Y-axis and circle size does not help here. I did not see any data in the Supplementary Information that could help with this. Perhaps I’m missing something due to not being so familiar with network analyses but whatever I’m missing could also be missed by other readers. So, further explanation would be helpful.”

RESPONSE: Thank you for noting that this part of the text and figure was unclear. To address this, we have first and foremost revised entirely Figure 1 to more clearly display these results: our updated line graph clearly shows that P6 scores the highest (but we agree with the reviewer that other nodes, like P3 and P12, score high as well, and this is now better visualized). Second, we have included a caption underneath Figure 1 to explain what each panel illustrates (for example, the P17 node now can more clearly be seen to be highest for the right-wing population). Finally, we have also updated parts of our results section to more clearly explain these patterns. We hope these major changes clarify that what these centrality scores indicate.

Finally, the reviewer says, “In Table 2, we are presented with “predictability indices ... that are highest uniquely for each network.” I understand the general approach but do not know if there is a convention in network analysis for deciding on these or whether the authors had their own specific criteria. This could be clarified.”

RESPONSE: There are no guidelines for choosing the highest predictability index. However, this result is purely descriptive and not meant to inferentially adjudicate clear differences across the three populations (as centrality indices, for instance, were meant to do). We now clarify these points in our interpretation and outline of results.

As I hope will be obvious, we have gone to some considerable efforts to revise our manuscript in the light of the comments of both reviewers, and we believe that the resulting paper is both stronger and clearer as a consequence. We hope that it is now suitable for publication in PLoS One.

---

## [Decision Letter · Decision Letter 1]

7 Aug 2025

Dear Dr. Bentall,

We look forward to receiving your revised manuscript.

Kind regards,

Cengiz Erisen

Academic Editor

PLOS ONE

Journal Requirements:

Reviewers' comments:

Reviewer's Responses to Questions

**Comments to the Author**

Reviewer #1: All comments have been addressed

Reviewer #3: All comments have been addressed

2. Is the manuscript technically sound, and do the data support the conclusions?

Reviewer #1: Yes

Reviewer #3: Yes

3. Has the statistical analysis been performed appropriately and rigorously?

Reviewer #1: Yes

Reviewer #3: Yes

4. Have the authors made all data underlying the findings in their manuscript fully available?

Reviewer #1: Yes

Reviewer #3: Yes

5. Is the manuscript presented in an intelligible fashion and written in standard English?

Reviewer #1: Yes

Reviewer #3: Yes

Reviewer #1: I appreciate the authors taking my comments seriously. I am satisfied with the revisions (including their responses to R2's comments).

Reviewer #3: The study offers a timely and methodologically sophisticated contribution to the literature on political belief systems by reviving and empirically testing Converse’s account using psychometric network analysis. Clearly, you have carefully and constructively engaged with prior reviewer feedback, and the manuscript is substantially improved as a result.

Strengths

The core contribution of the paper lies in its empirical demonstration that belief systems among left- and right-wing individuals are more interconnected—i.e., more “constrained”—than those among centrists, consistent with Converse’s theoretical expectations. This finding is both intuitive and consequential, and your use of Gaussian Graphical Models (GGMs), network comparison tests, and predictability indices is methodologically sound and well-motivated.

Your attention to robustness is commendable: the use of alternative cutoffs for defining centrism, random subsampling, and multiple estimation techniques helps confirm the stability of your findings. I also found the replication of community structure across ideological groups compelling. The emergence of three consistent clusters—mapped onto right-wing authoritarianism, altruism/cooperativeness, and personal liberalism—adds interpretive depth beyond mere global strength comparisons.

The discussion is well-organized and appropriately cautious in most places. Your removal of the term “woke” and refinement of cluster labels in response to reviewer comments improved the clarity and neutrality of your argumentation.

Points for Further Improvement

• Overstatement of Novelty

The manuscript at times overemphasizes its theoretical innovation. Prior studies (e.g., Dalege et al., 2017, 2018; Boutyline, 2017) have already used relational or network-based methods to analyze political belief structure and ideological consistency. Your contribution is essential in comparative and methodological replication, but it should be framed as a confirmation, extension, or refinement of existing findings rather than a novel theoretical departure.

• Imbalance in Issue Content

The set of 18 political issues used to construct the networks skews heavily toward social and cultural issues. By your count, only four items plausibly pertain to economic conservatism. This asymmetry likely influences the factor structure and centrality findings (e.g., the prominence of gay and trans rights in network centrality). While you acknowledge this in the discussion, the potential implications for generalizability and interpretation should be developed further. For instance, would the same triadic structure emerge if the item pool were more balanced?

• Reliance on Self-Reported Ideology

The left-centre-right groupings are based on a single-item self-placement measure, which introduces interpretive uncertainty. Respondents may self-label in ways that do not correspond to their policy views. A brief validation—e.g., correlating self-placement with average position on left-right-sorted items—would help reassure readers of the internal coherence of groupings.

• Childrearing Scale Use and Reliability

The childrearing preferences scale validates the factor structure, but its internal consistency is modest (α = .58). While you note this, the discussion still leans on it to support external validity. It may be helpful to either downplay this component or consider adding a robustness check using a different authoritarianism proxy, if available.

• Figure Interpretation and Accessibility

The revised figures are improved, but still dense. Readers unfamiliar with network visualizations may struggle to understand node predictability or strength centrality intuitively. Supplementing the main figures with simplified annotated versions in the appendix (e.g., highlighting a few example nodes and their strongest links) would aid interpretation.

• Terminology and Cluster Labels

The revised cluster labels (“right-wing authoritarianism,” “altruism and cooperativeness,” “personal liberalism”) are an improvement, but still feel loosely defined. Especially for “altruism and cooperativeness,” you may wish to tie the cluster more explicitly to existing ideological constructs—e.g., cosmopolitanism, universalism, or social solidarity—and clarify whether these clusters are assumed to be substantive dimensions or simply data-driven groupings.

• Interpretation of Central Nodes

The finding that gay rights (P6) is the most central node in the left and centre networks is intriguing. However, the manuscript does not theorize why this might be the case. If this node acts as a “bridge” across communities, or is ideologically symbolic, this deserves brief discussion. Similarly, the prominence of “public demonstrations” in the right-wing network could be interpreted more sharply—perhaps as a marker of anti-protest or status-quo attitudes.

This well-executed and thoughtful study will interest scholars working on political psychology, ideology, and beliefs. It carefully operationalizes a classic theoretical claim using modern methods, and the authors have responded thoroughly to prior critiques. The manuscript is suitable for publication with modest clarifications and framing adjustments—especially regarding the scope of its contribution and the implications of its item pool.

**Do you want your identity to be public for this peer review?** For information about this choice, including consent withdrawal, please see our Privacy Policy

Reviewer #1: No

Reviewer #3: No

---

## [Author Response · Author response to Decision Letter 2]

30 Aug 2025

The authors would like to thank the two reviewers for their further diligence in reviewing our manuscript. We note that both referees have answered ‘yes’ to all the mandatory questions about the suitability of our paper for publication, including question 1 about our response to previous feedback (“All comments have been addressed). Reviewer #1 furthermore says, “I appreciate the authors taking my comments seriously. I am satisfied with the revisions”; Reviewer #2 says, “Clearly, you have carefully and constructively engaged with prior reviewer feedback, and the manuscript is substantially improved as a result” but requests some further, additional changes, as follows:

1. The manuscript at times overemphasizes its theoretical innovation.

The referee says, “Prior studies (e.g., Dalege et al., 2017, 2018; Boutyline, 2017) have already used relational or network-based methods to analyse political belief structure and ideological consistency. Your contribution is essential in comparative and methodological replication, but it should be framed as a confirmation, extension, or refinement of existing findings rather than a novel theoretical departure.`’

We have already acknowledged the previous network studies by Dalege, Boutyline, and Brandt. Dalege’s work focused on the prediction of voting behaviour, and both Boutyline and Brandt included political identification as nodes within their networks. Our unique contribution has been to show that meaningful networks emerge purely from the organization of policy preferences (without symbolic nodes) and that, when people with different ideological identifications are compared, meaningful differences then emerge in network structures that are consistent with Converse’s theory.

In order to satisfy the reviewer and yet still highlight our unique approach, we have revised the beginning on page 28 of the revised manuscript of our discussion so that it now says, “This work extends previous research that has applied network models to analyse political beliefs, for example by Dalege and colleagues (32-34), Boutyline and colleagues (35) and Brandt and colleagues (36). However, the nodes in our networks were restricted to attitudes towards a range of policies. In contrast to previous analyses, we deliberately chose not to include behavioural measures such as voting history (33) or symbolic demonstrations of political affiliation such as party membership (35) because……”

2. Imbalance in issue content

The referee says, “The set of 18 political issues used to construct the networks skews heavily toward social and cultural issues. By your count, only four items plausibly pertain to economic conservatism. This asymmetry likely influences the factor structure and centrality findings (e.g., the prominence of gay and trans rights in network centrality). While you acknowledge this in the discussion, the potential implications for generalizability and interpretation should be developed further. For instance, would the same triadic structure emerge if the item pool were more balanced?”

Given that we have already acknowledged this limitation in the discussion (as the reviewer acknowledges), we do not see how we can do much more about this. However, we have added a further comment about this to the discussion on page 32 of the revised manuscript, where we now say: “It is therefore possible that a different structure would have emerged if we had included more economic items, perhaps with a specific community relating to economic beliefs, and perhaps reducing the influence of conservative items, for example relating to gay and trans rights”.

3. Reliance on self-reported ideology

The reviewer says, “The left-centre-right groupings are based on a single-item self-placement measure, which introduces interpretive uncertainty. Respondents may self-label in ways that do not correspond to their policy views. A brief validation—e.g., correlating self-placement with average position on left-right-sorted items—would help reassure readers of the internal coherence of groupings.”

We have already shown that the three self-defined ideological groups differ significantly on all 18 items in exactly the manner expected. Hence, this correlation is redundant. Nonetheless, out of respect for the referee we have calculated the correlation as requested, which is .504, p < .001, which we have now added on page 20 of the revised manuscript, and included in our discussion on page 33.

4. Childrearing scale use and reliability

The reviewer says: “The childrearing preferences scale validates the factor structure, but its internal consistency is modest (α = .58). While you note this, the discussion still leans on it to support external validity. It may be helpful to either downplay this component or consider adding a robustness check using a different authoritarianism proxy, if available.”

No additional robustness checks are available with the current dataset, which did not include additional measures of authoritarianism. We have therefore added the following comment to the discussion on page 33: “…. this finding should be treated with caution and future studies might consider a wider range of measures to validate observed community structures of political beliefs.

5. Figure interpretation and accessibility

The reviewer says: “The revised figures are improved, but still dense. Readers unfamiliar with network visualizations may struggle to understand node predictability or strength centrality intuitively. Supplementing the main figures with simplified annotated versions in the appendix (e.g., highlighting a few example nodes and their strongest links) would aid interpretation.”

We appreciate the continued attention to figure clarity. However, we have aligned our figures with established conventions in the network literature and have previously devoted considerable effort to revising them to be consistent with the last round of reviewer feedback.

We appreciate that network visualizations may be unfamiliar to researchers in some fields but within political science, and psychology, more broadly, these representations are intuitive and familiar to most scholars. We therefore believe the current presentation strikes an appropriate balance between precision and interpretability.

For these reasons, and to maintain clarity without unnecessary redundancy, we have kept the figures as they are without adding any additional supplementary versions of them (which may confuse rather than enhance interpretability).

6. Terminology and cluster labels

The referee says, “The revised cluster labels (“right-wing authoritarianism,” “altruism and cooperativeness,” “personal liberalism”) are an improvement, but still feel loosely defined. Especially for “altruism and cooperativeness,” you may wish to tie the cluster more explicitly to existing ideological constructs—e.g., cosmopolitanism, universalism, or social solidarity—and clarify whether these clusters are assumed to be substantive dimensions or simply data-driven groupings.”

Again, we have already responded to the reviewer’s advice on this. The choice of labels is inherently subjective and could be debated further without resolution. We therefore do not propose to revise our labels further.

7. Interpretation of central nodes

The referee says: “The finding that gay rights (P6) is the most central node in the left and centre networks is intriguing. However, the manuscript does not theorize why this might be the case. If this node acts as a “bridge” across communities, or is ideologically symbolic, this deserves brief discussion. Similarly, the prominence of “public demonstrations” in the right-wing network could be interpreted more sharply—perhaps as a marker of anti-protest or status-quo attitudes.”

We did in fact theorise why gay rights might have been a very central node, pointing to past research that shows that attitudes towards sexual minorities seems to be related to disgust sensitivity, which is a well-replicated predictor of social conservativism. Hence, we are not proposing that there is anything ideologically symbolic about beliefs about sexual minorities; it is related to a fundamental psychobiological process. We could add more about this but, to do the issue justice would significantly lengthen the manuscript and we think that, in this context, it is sufficient to point the reader towards the relevant literature. To this end, we have added a further reference to the work of Haidt on page 31 of the revised manuscript.

We have also added a brief paragraph on page 31 to elaborate on the interpretation of the ‘public demonstrations’ node as suggested by the reviewer.

We went to very considerable lengths to revise the manuscript in the light of both referees’ comments after the last iteration of reviews, and both referees expressed satisfaction with those efforts. As I hope will be obvious, we have now gone to further efforts to revise our manuscript in the light of the most recent comments of reviewer #2. We do not see much prospect of improving the manuscript further after such a thorough and lengthy process of review and revision. We hope that you will now be able to agree that the resulting paper is suitable for publication in PLoS One.

---

## [Decision Letter · Decision Letter 2]

16 Sep 2025

The structure of mass political belief systems: A network approach to understanding the left-right spectrum

PONE-D-24-34416R2

Dear Dr. Bentall,

We’re pleased to inform you that your manuscript has been judged scientifically suitable for publication and will be formally accepted for publication once it meets all outstanding technical requirements.

Kind regards,

Cengiz Erisen

Academic Editor

PLOS ONE

Additional Editor Comments (optional):

Reviewer #3:

Reviewers' comments:

Reviewer's Responses to Questions

**Comments to the Author**

Reviewer #3: All comments have been addressed

2. Is the manuscript technically sound, and do the data support the conclusions?

Reviewer #3: Yes

3. Has the statistical analysis been performed appropriately and rigorously?

Reviewer #3: Yes

4. Have the authors made all data underlying the findings in their manuscript fully available?

Reviewer #3: Yes

5. Is the manuscript presented in an intelligible fashion and written in standard English?

Reviewer #3: Yes

Reviewer #3: I appreciate the authors’ careful and thorough revisions. The manuscript has improved substantially in clarity, coherence, and scholarly contribution. The responses to the reviewers’ comments are convincing, and the revisions to the analyses and argumentation strengthen the paper considerably. The manuscript now reads as a polished and persuasive piece of work that makes a meaningful contribution to the field.

**Do you want your identity to be public for this peer review?** For information about this choice, including consent withdrawal, please see our Privacy Policy

Reviewer #3: No

---

## [Editor Report · Acceptance letter]

PONE-D-24-34416R2

PLOS ONE

Dear Dr. Bentall,

I'm pleased to inform you that your manuscript has been deemed suitable for publication in PLOS ONE. Congratulations! Your manuscript is now being handed over to our production team.

Kind regards,

on behalf of

Dr. Cengiz Erisen

Academic Editor

PLOS ONE